# Simulation and Experimental Analysis of Surface Defects in Turning of TiC_p_/TC4 Composites

**DOI:** 10.3390/mi14010069

**Published:** 2022-12-27

**Authors:** Haixiang Huan, Chilei Zhu, Biao Zhao, Wenqiang Xu, Ke Zhang

**Affiliations:** 1School of Mechanical Engineering, Yancheng Institute of Technology, Yancheng 224051, China; 2College of Mechanical and Electrical Engineering, Nanjing University of Aeronautics and Astronautics, Nanjing 210016, China

**Keywords:** TiC_p_/TC4 composites, particle debonding, microcracks, surface defects

## Abstract

Processing TiCp/TC4 composites has always been difficult due to the mismatch between the mechanical and thermal properties of the matrix and the reinforced particles, which results in a variety of machined surface defects. To expose the mechanism of defect generated on the cutting surface of TiCp/TC4 composites and improve their cutting surface quality, a 3D finite element orthogonal turning simulation model of TiCp/TC4 composites is developed. The failure at the matrix-particle interface and the fracture and removal mechanism of the reinforcing phase particles are analyzed from a microscopic perspective using a single particle cutting simulation model. In addition, a three-dimensional cutting simulation model with randomly dispersed TiC particles and a volume fraction of 5% is developed, and various forms of cutting surface defects of TiCp/TC4 composites are examined. To verify the validity of the finite element simulation model, TiCp/TC4 composites with a volume fraction of 5% are selected for turning tests. For various cutting tools and particle relative positions, the simulation and test results show that the removal of particles takes the following forms: debonding, crushing, brittle fracture, and slight fracture at the top, leading to a shallow cavity, microcracks, residual TiC particles embedded in the cavity, and surface defects with severe plastic deformation of the matrix surrounding the cavity on the machined surface.

## 1. Introduction

Particle Reinforced Metal Matrix Composites (PRMMCs) consist of matrix metal and reinforcement particles. The particles of the reinforced phase are often composed of titanium carbide or alumina, while the matrix material is typically composed of aluminum, titanium, and magnesium. The manufacturing costs for this material are decreased, the process is easy, and typical methods, such as rolling and forging, can be employed for further processing. A material that is macroscopically isotropic is a promising innovation. Particle-reinforced titanium matrix composites (PTMCs) are used extensively in aerospace, automotive, and electrical applications [1,2]. On the machined surface of particle-reinforced composites, the inclusion of high-hardness reinforcing phases causes defects such as microcracks, shallow cavities, particle fracture, matrix tearing, matrix hardening, and scratches. These issues are significant obstacles to the widespread adoption of PTMCs. In order to promote the application of PTMCs, it is crucial to investigate the effect of the particle fracture process on the production of machined surface defects in PTMCs.

The addition of reinforcement particles has resulted in PRMMCs being machined with poor machined surface finish and numerous forms of surface flaws [3]. On the machined surface of in situ aluminum matrix composites, the flaws such as pits, voids, microcracks, protrusions, matrix smears, matrix tears and feed marks are identified during cutting experiments conducted by Xiong et al. [4]. Qiao et al. [5] conducted performed single-factor turning experiments on A03190/304 composite material in a single-factor turning test. Reducing the feed rate and depth of cut, or raising the cutting speed, is shown to reduce machined surface flaws. Huo et al. [6] conducted a significant number of turning tests on PTMCs and discovered that the machined surface roughness Ra could be kept below 0.5 μm using cutting parameters in the range of v = 80~100 m/min, ap = 0.30~0.60 mm, and f = 0.006~0.10 mm/r.

Finite element simulation is a useful and effective methodology for studying the cutting process. Cutting simulation has several advantages, such as making it simple to monitor the micro-cutting process, collecting data on cutting force heat immediately, and conserving materials, et al. [7]. Combining simulation and testing, Elkaseer et al. [8] investigated the surface creation during the precise turning of 316L stainless steel. Chen et al. [9] established a macroscopic equivalent homogeneous material 3D finite element cutting simulation model based on quasi-static and SHPB tests to describe the machining process of SiCp/Al6063/30P composites. The error between the simulated cutting forces and experimental results was within 20%. Nonetheless, the analogous homogeneous model was unable to study the removal of material at the microstructure level.

It is possible to model the impact of particle and interfacial layer failure on the particle-cutting process by simulating single particle-cutting operations for PRMMCs. For SiCp/Al particle-reinforced composites with a body fraction ratio of 45%, Liu et al. [10] created a single-particle cutting model. When the tool cut the upper portion of the particle, the particle fractured brittlely, and the depth of surface defects decreased as the cutting speed increased; when the tool cut the lower portion of the particle, the removal of the particle changed from being pulled out to be crushed as the cutting speed increased. There are two mechanisms for particle breaking, according to Wu et al. [11]. One is caused by the cutting edge making direct contact with the particle, while the other is caused by the tool and particle interacting indirectly with the matrix. Laghari et al. [12] quantitatively examined the impact of interfacial characteristics on the strengthening effect of SiCp/Al composites by adding an interfacial layer to a single particle cutting simulation model. It is demonstrated that the stiff interface model had a stronger enhancing impact than the flexible interface model as the thickness of the interface increased. When simulating the cutting of particle-reinforced composites using finite elements, Ghandehariun et al. [13] and Meng et al. [14] both added viscous domain units. In order to analyze the matrix deformation and the interactions between tool particles during machining, Pramanik et al. [15] employed a finite element method, in which physical processes such as wear, particle debonding, displacement, and non-uniform deformation of the matrix material are investigated.

Zhang et al. [16] found that for SiC particle sizes between 10 and 30 μm, the principal cutting force decreased as the cutting speed increased; Throughout the cutting process, the contact area between the tool’s front edge surface and the second deformation zone frequently exhibited the greatest equivalent force. In order to increase the accuracy of the cutting model, Wang et al. [17] analyzed the surface contours machined by turning tests and formulated the size of the particles in the cutting model based on the crater depth of the turning surface. Huan et al. [18] studied the impact of the relative position relationship between the tool and the particles in the form of TiC particle removal while including particles with three different cutting depths in the same model to prevent boundary conditions from impacting simulation findings, the findings of which showed that the errors in cutting temperatures and forces are within 15%.

Numerous researchers have investigated increased particle random distribution cutting simulation models in order to increase the realism of cutting models. In order to investigate the primary formation mechanisms of surface defects in SiCp/Al composites at high-volume fractions, Wang et al. [19] developed randomly distributed circular SiC particle models and polygonal SiC particle models. The results showed that SiC rotation, pull-out, large unraveling, microfracture, and cut-through are the primary mechanisms of surface defect formation. Wu et al. [20] evaluated 2D and 3D cutting simulation models and discovered that while the 2D model could reflect some aspects of material removal, it had limited ability to forecast microstructural changes. Fan et al. [21] created a cutting simulation model with a random distribution of polyhedral SiC particles. The cutting surface shape is shown to be significantly influenced by the SiC particle failure mode.

In conclusion, referring to the previous works, the formation of surface defects on PRMMCs using simulation techniques and tests is studied. During the processing of PRMMCs, the mechanism of particle shattering and surface creation is discovered. Analyses are carried out on the interactions between tool-particle, tool-matrix, and matrix-particle throughout the machining process. Combining SEM results and using programming languages to improve the accuracy of cutting models. 

However, there are only two-dimensional cutting simulation studies for PTMCs, and the simulation models lack the interface between the particles and the substrate. In order to explore the material removal during the processing of PTMCs and to overcome the shortcomings of the 2D orthogonal cutting simulation model, this paper simplifies the TiC particles into spheres and constructs a 3D microstructure-based orthogonal cutting simulation model. To study the formation process of brittle fracture, debonding, and particle fracture, an effort to disclose the forming mechanism of surface defects in TiCp/TC4 composites is performed.

## 2. Finite Element Modeling and Test Conditions

### 2.1. Finite Element Modeling

This study employs the Software package Abaqus to create a three-dimensional microscopic model based on the Lagrange model in order to investigate the creation mechanism of the cutting surface of high-speed cutting titanium matrix composites. Figure 1 illustrates the material employed in this study, which is composed of 5% reinforcement TiCp/TC4 composites. The matrix material is TC4, while the hard particles are TiC.

#### 2.1.1. Establishment of the Constitutive Matrix Model

An excellent representation of metal can be found in the Johnson-Cook (J-C) constitutive model, which is an equation relating stress, strain, and temperature [22]. It is assumed that the strain, strain rate, and temperature all have independent effects on the TC4 matrix throughout the cutting process. Its plastic behavior is described by the J-C constitutive model, and the constitutive equation is as follows [23]:(1)$$\sigma =\left[A+B\varepsilon^{n}\right]\left[1+C\ln\left(\frac{\overset{\cdot }{\varepsilon }}{\overset{\cdot }{\varepsilon_{0}}}\right)\right]\left[1-{\left(\frac{T-T_{r}}{T_{m}-T_{r}}\right)}^{m}\right]$$
where *A* represents the yield strength under quasi-static conditions; *B* represents the strain hardening parameter; $\varepsilon_{p}$ represents Equivalent Plastic Strain; *n* represents the hardening index; *C* represents the strain rate strengthening parameter; $\dot{\varepsilon }$ represents the equivalent plastic strain rate; $\dot{\varepsilon_{0}}$ represents reference strain rate for the materials; *T_r_* stands for the normal temperature coefficient, usually 25 °C; $T_{m}$ stands for the melting point of the material; *m* stands for the thermal softening coefficient. The J-C constitutive model parameters used for the matrix material are shown in Table 1.

The J-C shear fracture criterion is applied to TC4 as the chip separation criterion to replicate the cutting formation. The element’s failure is determined using the corresponding plastic strain *ε*. The basis for determining if a material has failed is the damage parameter $\omega_{s}$, which gradually rises with the plastic deformation of the material. The damage parameter $\;\omega_{s}$ expression is:(2)$$\omega_{s}=\sum \frac{{\Delta \varepsilon }^{pl}}{{\overset{-}{\varepsilon }}_{0}^{pl}}$$

When $\omega_{s}$ = 1, the material starts to fracture, and the chips start to separate from the workpiece, whereas $\Delta \varepsilon^{pl}$ is the corresponding plastic strain increment during the process of material deformation; ${\overset{-}{\varepsilon }}_{0}^{pl}$ is the analogous strain when the material first fails. In this case, ${\overset{-}{\varepsilon }}_{0}^{pl}$ can be acquired by
(3)$${\overset{-}{\varepsilon }}_{0}^{pl}=\left(d_{1}{+d}_{2}e^{d_{3}\sigma^{*}}\right)\left({1+d}_{4}\ln\frac{\overset{g}{\varepsilon^{pl}}}{\overset{g}{\varepsilon_{0}}}\right)\left[1-d_{5}\left(\frac{T-T_{r}}{T_{m}-T_{r}}\right)\right]$$
where $d_{1}$~$d_{5}$ is the coefficient parameter of the J-C fracture failure criterion model. It can be seen from the formula that the stress $\sigma^{*}$, strain rate $u_{n0}=\frac{2G_{f}^{I}}{\sigma_{tu}^{I}}$, and temperature *T* of the material determines the equivalent strain when the material fails. In this work, the fracture failure criterion model coefficient parameters ($d_{1}$~$d_{5}$) of TC4 titanium alloy are shown in Table 2.

#### 2.1.2. Establishment of the Constitutive Particle Model

TiC particles have greater rigidity and stiffness than the TC4 matrix and fracture according to the generalized HooK law, which is a totally linear material. TiC particles seldom undergo plastic deformation after undergoing elastic deformation but directly cause fissures. Since TiC particles are typical brittle materials, they are modeled as isotropic linear elastics [25].

To mimic the brittle fracture of the TiC particles, a brittle fracture damage model has been developed. The vertical stress criteria are utilized to detect particle fracture beginning. The vertical stress criteria are derived from the formula provided in [25]:(4)$$\max(\sigma_{1},\sigma_{2},\sigma_{3})=\sigma_{0}$$
where $\sigma_{0}$ is the tensile strength of the TiC particles and *σ*_1_, *σ*_2_, and *σ*_3_ are the main stress components.

After the brittle fracture of a TiC particle, the failure evolution is governed by the fracture energy criterion, and the formula for normal crack displacement is as follows:(5)$$u_{n0}=\frac{2G_{f}^{I}}{\sigma_{tu}^{I}}$$
where $G_{f}^{I}$ is the type I material’s fracture energy, $\sigma_{tu}^{I}$ is the failure stress, and $u_{n0}$ is the normal displacement at failure.

The shear retention model is constructed as a function of crack opening strain in order to characterize the shear stress-induced damage development. The shear modulus of a crack is computed using the following formula:(6)$$G_{C}=\rho (e_{nm}^{ck})G$$
where *G* represents the shear modulus of the undamaged material is the cracking strain; $G_{c}=\rho \left(e_{nm}^{ck}\right)$ is the shear retention factor. The formula is as follows:(7)$$\rho (e_{nm}^{ck})={\left(1-\frac{e_{nm}^{ck}}{e_{max}^{ck}}\right)}^{p}$$
where $e_{nm}^{ck}$ is the crack opening fission, and p and $e_{max}^{ck}$ are the material parameters.

Table 3 shows the relevant parameters required for the brittle fracture failure of TiC particles. 

#### 2.1.3. Establishment of Interface Cohesion Model

The secondary stress failure criterion is used in this study to determine the initiation of adhesive interface damage. When the quadratic interaction function involving the nominal stress ratio reaches 1, the damage is assumed to begin. The criterion could be expressed as [14]
(8)$${\left(\frac{\langle t_{n}\rangle }{t_{n}^{0}}\right)}^{2}+{\left(\frac{t_{1}}{t_{1}^{0}}\right)}^{2}+\left(\frac{t_{2}}{t_{2}^{0}}\right)=1$$
where 〈 〉 is the McCauley bracket, defined as When *x* ≥ 0, ⟨*x*⟩ = *x* and when *x* < 0, ⟨*x*⟩ = 0; *t*
$t_{i}^{0}$ is the interface strength component (*I* = *n*, 1, 2). To calculate conveniently, the linear damage evolution law based on effective displacement is selected. The damage associated with the failure mechanism is represented by the elastic stiffness degradation $D_{i}$ in the damage evolution stage, which varies from 0 to 1. According to the law of damage evolution [14]:(9)$$D_{i}=\frac{\delta_{i}^{F}(\delta_{i}^{max}-\delta_{i}^{0})}{\delta_{i}^{max}(\delta_{i}^{F}-\delta_{i}^{0})}$$
where $\delta_{i}^{0}=t_{i}^{0}/K_{ii}$ is the initial displacement of damage, and $\delta_{i}^{F}=2G_{i}^{f}/t_{i}^{0}$ is the failure displacement.
(10)$${\left(\frac{G_{n}}{G_{n}^{f}}\right)}^{\alpha }+{\left(\frac{G_{1}}{G_{1}^{f}}\right)}^{\alpha }+{\left(\frac{G_{2}}{G_{2}^{f}}\right)}^{\alpha }=1$$
where the fracture energy component $G_{i}^{f}\left(i=n,1,2\right)$ and index are both greater than 0. The parameter *α* = 1 from the reference is used in this paper. The cohesive elements will be eliminated when the damage development progresses, and the average interface damage reaches its maximum level. The complete fracture and separation of mock components can be accomplished using the element removal method in most cases. There will be no resistance to the separation of the matrix from the interface once the cohesive unit has been removed.

#### 2.1.4. Grid and Boundary Conditions

The TiC particles are divided into eight sections along the center datum, and the particle mesh type is C3D8R, an eight-node, linear, reduced-integration hexahedral cell, as illustrated in Figure 2 in order to achieve an ideal cell mesh for any model region affecting the analysis of the results. At the particle-substrate interface, a zero-thickness COH3D8, an eight-node cohesive cell, is injected.

As depicted in Figure 3, since the tool is fixed in the y and z directions and the workpiece is fixed in the *x*, *y*, and *z* directions on its underside, the tool can move in the *x* direction at a cutting speed Vc of 45 m/min.

### 2.2. Conditions and Methods of Cutting Test

#### 2.2.1. Materials and Conditions

The matrix material of TiC_p_/TC4 composites used in the experiment is TC4, which is prepared by the in situ formation method, and TiC is a hard reinforcing phase. Figure 1 displays the SEM picture of the 5% TiCp/TC4 composite, while Table 4 displays the material’s mechanical and physical characteristics.

The machine tool used in this test is NYC-6140 numerically controlled machine tool, as shown in Figure 4.

The PCD tool is welded by a PCD polymer block and cemented carbide substrate. For the PCD tool assembly, the tool rod model is PSSNR2020K12. According to Figure 5, the PCD tools [28] selected are a mixture of 30 and 2 μm diamonds. Table 5 displays the specific PCD tool parameters.

#### 2.2.2. Test Scheme

The goal of this experiment is to investigate the change rule of the PCD tool in turning TiC_p_/TC4 composite surface defect forming. The table below, Table 6, lists the specific test parameters.

#### 2.2.3. Test Observation Equipment

With the purpose of obtaining SEM images of actual particle breakage and surface defects, the TiCp/TC4 composites are observed in the Nova Nano SEM 450 after turning. Figure 6 depicts the equipment used for observation.

## 3. Single Particle Cutting Simulation Results and Testing Analysis

TiC particle is one of the primary elements influencing the formation of surface flaws in TiCp/TC4 composites during turning [29]. To explore the removal mechanism of TiC particles during the turning process, only a single TiC particle is incorporated into the model. The relative position relationship between TiC particles and PCD tools is investigated under three typical conditions. As depicted in Figure 7, the center of the TiC particle and the center of the tooltip are on the same horizontal line; the center of the TiC particle is below and above the center of the tooltip, respectively.

### 3.1. Cutting the Middle of the Particle

As depicted in Figure 8, the particle’s center and the tool’s center are placed on the same horizontal line. As depicted in Figure 8a, when the chip rolls up the rake face, the matrix-particle interface of the particles at the chip side breaks, and the TiC particles begin to crack. As illustrated in Figure 8b, when the tool comes into contact with the particles, the tension on the particles increases dramatically, the upper and lower sides of the TiC particles are crushed by the tool, and the area of particles that have fractured increases. As shown in Figure 8c, as the tool is cut further, the fractures develop on both sides of the maximum stress of TiC particles, and brittle fractures emerge along the crack particles to form a shallow hollow. When the tool presses particle fragments into the matrix, the fragments generate a strong compressive stress. As depicted in Figure 8d, these stresses interact with the matrix, causing fissures between the particles and the matrix. As illustrated in Figure 8e, due to the uneven particle brittle fracture, some TiC particles are lodged in the matrix, while some TiC particle fragments are forced into the chip to produce the shallow cavity. Figure 8f depicts the fragmentation of particles in the center of a cutting particle when Vc = 45 m/min. The test outcomes resemble Figure 8e. Under the action of tool extrusion, TiC particle fragments damage the left matrix, generating a shallow hollow on the machined surface.

As shown in Figure 8a, as the tool approaches the particles, the stress on the particles increases from 500 MPa to 3000 MPa. As depicted in Figure 8b, the brittle fracture of the particles occurs between 4000 and 8000 MPa. As depicted in Figure 8c, with the brittle fracture of large-area particles, the stress on the particles embedded in the matrix decreases to less than 3700 MPa, and as depicted in Figure 8d, the compressive stress on the failed particles is between 2000 and 4500 MPa.

### 3.2. Cutting the Upper Part of the Particle

As shown in Figure 9, the particle’s center is below the tool’s center, and the bulk of particles are below the cutting path. As illustrated in Figure 9a, since the embedded TiC particles hinder the flow of the ductile matrix, the high tensile stress during chip formation is focused above the matrix-particle interface. As depicted in Figure 9b, the chip formation increases the plastic flow of the matrix near the matrix-particle interface, and the particle matrix-particle interface begins to fail. When a tool comes into touch with particles, the tool’s tension increases dramatically. Figure 9c reveals that the TiC particles begin to fracture when the primary stress approaches the limit of the brittle fracture criterion. As illustrated in Figure 9d, due to the extrusion of the knife side, the particles are gradually forced into the matrix, and a portion of the particles near the knife side is severed, therefore expanding the fracture area. A portion of the chopped particles is pressed into the matrix. As demonstrated in Figure 9e, the primary failure modes of the particles are cutting and tool extrusion, and the machined surface is rather flat. Figure 9f depicts the particle fragmentation that occurs when an instrument with a cutting speed of Vc = 45 m/min cuts the upper portion of particles. The experimental outcomes resemble Figure 9e. The TiC particles have a somewhat broken surface on top, while the machined surface is quite flat.

As shown in Figure 9a, as the tool approaches the particles, the stress of the particles remains below 500 MPa; as shown in Figure 9b, when the stress in the shear zone exceeds 1400 MPa, the matrix-particle interface begins to fail; as shown in Figure 9c, when the tool contacts the particles, the maximum stress on the particles reaches 8755 MPa; As shown in Figure 9d, the extrusion stress on the front face of the particle exceeds 6000 MPa and the particle fractures; the extrusion stress on the back face is below 5300 MPa, resulting in a flatter surface after machining.

### 3.3. Cutting the Lower Part of the Particle

As seen in Figure 10, the particle’s center is situated above the tool’s center. As a result of the tool being pressed into the matrix, the matrix on the left side of the tool produces plastic flow, which leads to the matrix-particle interface failure above TiC particles. As depicted in Figure 10a, the matrix-particle contact on the side of the first deformation zone begins to fail. When the tool touches particles, the tool’s tension increases dramatically. As illustrated in Figure 10b, as the hardness of the reinforced phase TiC particles is significantly larger than that of the TC4 matrix, the interface beneath the particles begins to fail, and the particles tend to debond. As illustrated in Figure 10c, as the tool continues to squeeze the particles, the bottom of the majority of the failed particles at the bond interface detaches from the matrix, and the particles break in the area of contact between the tool and the matrix. As demonstrated in Figure 10d, due to the substantial difference in hardness between the particles and the substrate, the particles scrape the machined surface and eventually press into the substrate throughout the dial-out process. As shown in Figure 10e, as the tool moves through the void left by the removed particles, the stress reduces fast, and some particles are pressed into the matrix to produce chips. Figure 10f depicts particle fragmentation when the tool cuts the lower portion of particles at a cutting speed of Vc = 45 m per minute. The test outcomes resemble Figure 10e. TiC particles entirely detach during the removal process, leaving a crater on the machined surface.

Figure 10a Matrix-particle interface ruptures when the stress in the zone between particles and the matrix rises above 1400 MPa, as shown in Figure 10b,c. When the tool makes contact with the particles, the stress on the particles quickly rises to 32,000 MPa, causing a large number of particles in the contact area to debond and fail. Compressive stresses of 3000–7000 MPa are caused by particle fragments, as seen in Figure 10d.

## 4. Analysis of Cutting Surface Defects of TiC_p_/TC4 Composite

The TiCp/TC4 composite consists of a high-strength TC4 matrix and high-strength TiC particles. The flaws on the machined surface are intricate and varied. Combining the aforesaid particle removal mechanism analysis with the random distribution model of TiC particles, the formation process of cutting surface defects in TiCp/TC4 composites is further investigated.

### 4.1. Construction of TiC Particle Random Distribution Simulation Model

The simulation approach simulated the actual form of TiC particles as a sphere. Create a circle in two dimensions using the following equation [30]:(11)$$r_{i}{=L}_{0}+\left({2\mu }_{i}-1\right){gL}_{1}$$
where $L_{0}$ represents the average radius, $L_{1}$ represents the amplitude, and $\mu_{i}$ represents a chance number between 0 and 1. The following are the coordinates for the vertices:(12)$$\left\{ \begin{array}{l} x_{i}=x_{0}+r_{i}\\ y_{i}=y_{0}+r_{i}\\ z_{i}=z_{0}+r_{i} \end{array}\right.$$

The following procedure is used to convert the particle’s initial position $\left(x_{i},y_{i},z_{i}\right)$ into the final vertex coordinates $\left(x_{i}^{'},y_{i}^{'},z_{i}^{'}\right)$:(13)$$\left\{ \begin{array}{l} x_{i}^{'}=x_{i}cos(\alpha_{y})+\left[z_{i}cos(\alpha_{x})+y_{i}sin(\alpha_{x})\right]sin(\alpha_{y})\\ y_{i}^{'}=y_{i}cos(\alpha_{x})-z_{i}sin(\alpha_{x})\\ z_{i}^{'}=\left[z_{i}cos(\alpha_{x})+y_{i}sin(\alpha_{x})\right]cos(\alpha_{y})-x_{i}sin(\alpha_{y}) \end{array}\right.$$
where $\alpha_{x}$ and $\alpha_{y}$ are the rotation angles on the *x*-axis and *y*-axis, respectively. 

All the created spherical particles are then randomly distributed throughout the matrix. A check for overlap is performed in order to prevent interference between these particles. To minimize the effect of model boundaries on numerical simulation and maintain high computational efficiency, the length, height, and breadth of the model are set to 100 μm, 25 μm, and 25 μm, respectively. Figure 11 depicts the simulation model for the random distribution of composite particles reinforced with TiCp/TC4. The volume fraction of the particles is 5.08%, with 16 particles present.

### 4.2. Surface Defect Analysis of Brittle Particle Fracture

As seen in Figure 12, the machined surface of the TiCp/TC4 composite reveals a cavity inlaid with particles. As depicted in Figure 8c, when TiC particles are in close proximity to or facing the tool path during the cutting operation, the stress on the particles increases dramatically. When the stress value reaches a particular limit, the TiC particle boundary begins to fracture brittlely, while the center of the particle remains embedded in the matrix, resulting in shallow pits and embedded particles on the machined surface. As demonstrated in Figure 8d and Figure 12b, when the particles are compressed by the tool, they generate high compressive stress, which acts on the matrix and causes matrix crack propagation near the shallow cavity.

### 4.3. Particle Debonding Surface Defect Analysis

Figure 13 displays a cavity formed on the machined surface of TiCp/TC4 composites as a result of particle debonding. When TiC particles are close to the tooltip, and above the cutting path, substantial compressive stress is applied to the matrix between the rake face and the TiC particles. The direction of compressive stress is orthogonal to the rake face. As seen in Figure 10c, as the tool advances, the effect of this compressive stress develops, causing the particles to slide and flip inside the matrix and separate from the TC4 matrix. During the debonding procedure, the particles scrape the substrate, leaving a cavity of the same size as the particle size on the machined surface and scuffs on the left side of the cavity. The tool feed presses partially failed particle pieces into the matrix, as seen in Figure 13b.

### 4.4. Analysis of Microcrack on Machining Surface

Figure 14 depicts the appearance of microcracks on the machined surface of TiCp/TC4 composites. During the cutting process, as depicted in Figure 14c, the TiC particles beneath the cutting path are completely pushed into the matrix, and brittle fracture occurs above the TiC particles under compressive stress. In addition, as a result of the ironing impact of the tool’s reverse side on the machined surface, the TC4 matrix and TiC particles will rebound more strongly, resulting in TiC particles protruding from the machined surface. As depicted in Figure 14a,b, microcracks occur on the machined surface as a result of the combined action of matrix and particles following a temperature decrease.

## 5. Conclusions

In this paper, three-dimensional orthogonal cutting simulation models of TiCp/TC4 composites with single particles and randomly distributed particles, as well as turning tests, are developed. The removal process of TiC particles and its effect on the forming mechanism of machined surface defects are explored under varying relative positions of cutting tools and particles. The following are the conclusions:

(1) Through the analysis of the turning test surface of TiC_p_/TC4 composite material, the removal form of TiC particles is mainly cutting, crushing, brittle fracture, and debonding, and the corresponding microcracks and cavity appear on the machined surface. Consistent with the cutting simulation surface defects, 3-D simulation has a certain accuracy.

(2) To examine the mechanism of surface defect formation in TiCp/TC4 composites, simulated studies categorize the positional interaction between the PCD tool and TiC particles into three groups. When the cutting tool cuts the upper portion of the particles, the particles are somewhat broken or crushed, the machining surface exhibits microcavities or microcracks, and the quality of the machined surface is optimal. Due to the uneven stress of the particles, when the tool is cut through the center of the particles, the machining surface appears to be a shallow cavity embedded with reinforced particles, cracks occur in the matrix around the cavity, and the machined surface is relatively poor. When the tool reaches the lowest portion of the particles, the machining surface resembles the particle size of the cavity, and the cavity side is marred with scratches, severely degrading the surface quality.

(3) When cutting the upper portion of the particles, a compressive stress of more than 5000 MPa is generated by the direct contact between the rake face and the particles, causing them to fracture slightly. When the PCD cuts through the middle of the particles, the unequal distribution of particle stress leads to the brittle fracture of particles in stress regions above 4000 MPa. When the tool comes into touch with the lower portion of the particles, the stress suddenly increases, and the particles become debonded.

In this paper, a three-dimensional simulation model of cut TiCp/TC4 composites is developed. The significance of the results is that the mechanism of surface defects in TiCp/TC4 composites can be explained. However, as the particle distribution of the cutting simulation model is random, the matrix is difficult to divide into a hexahedral structure, and a more accurate cutting simulation model can subsequently be developed. Cutting speed has an effect on the surface formation of particle-reinforced composites, and in this paper, only the case of 45 m/min is compared between simulated and experimental results. The effect of cutting speed on the formation of surface defects can continue to be investigated in the future.

## Figures and Tables

**Figure 1 micromachines-14-00069-f001:**
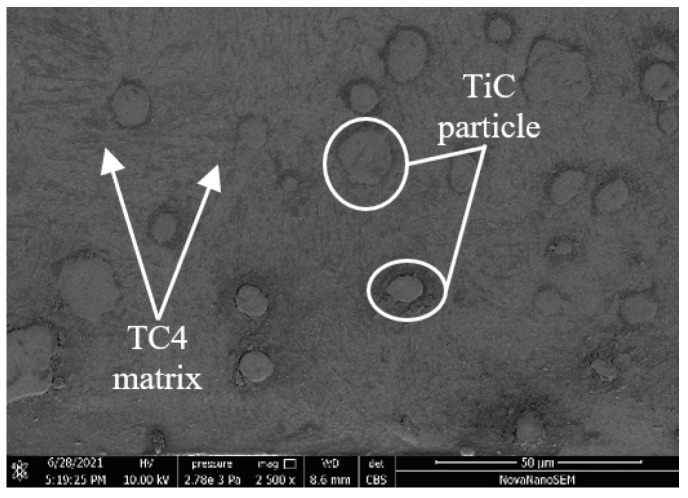
Scanning electron microscope images of 5% forged TiC_p_/TC4 composites.

**Figure 2 micromachines-14-00069-f002:**
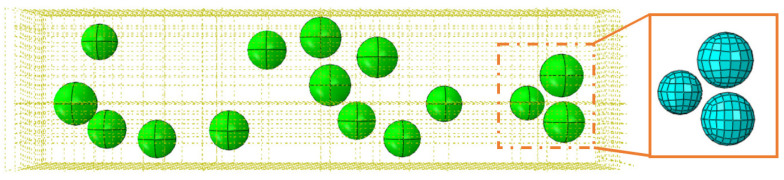
Diagram of TiC particle splitting.

**Figure 3 micromachines-14-00069-f003:**
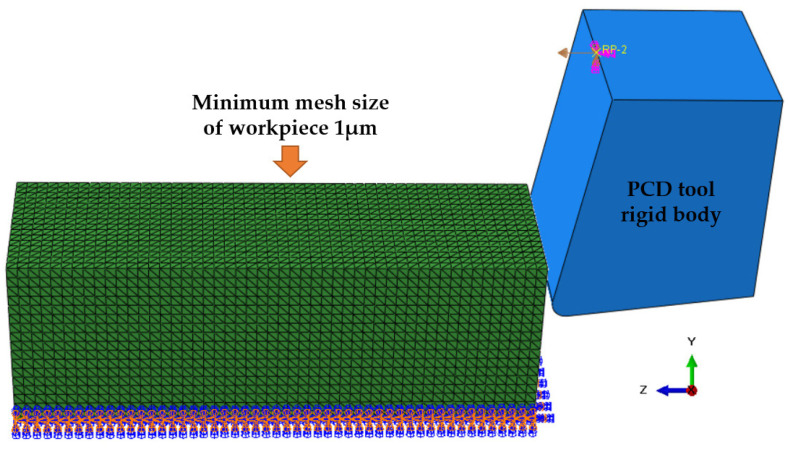
Diagram of boundary conditions of TiCp/TC4 composites.

**Figure 4 micromachines-14-00069-f004:**
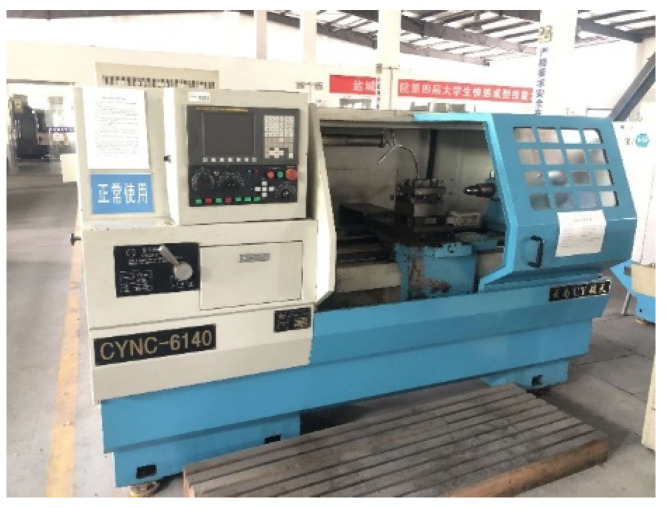
NYC-6140 numerically controlled machine tool.

**Figure 5 micromachines-14-00069-f005:**
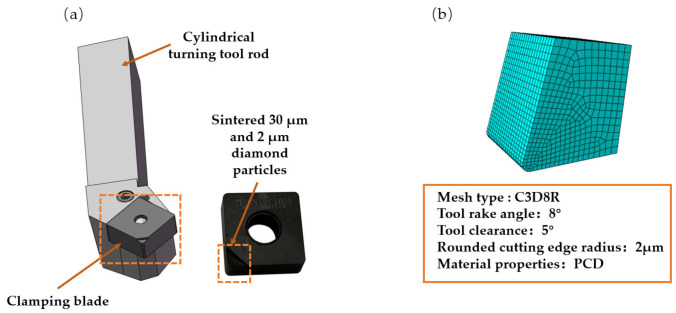
PCD tool schematic: (**a**) structure diagram of PCD tool and (**b**) PCD tool simulation model.

**Figure 6 micromachines-14-00069-f006:**
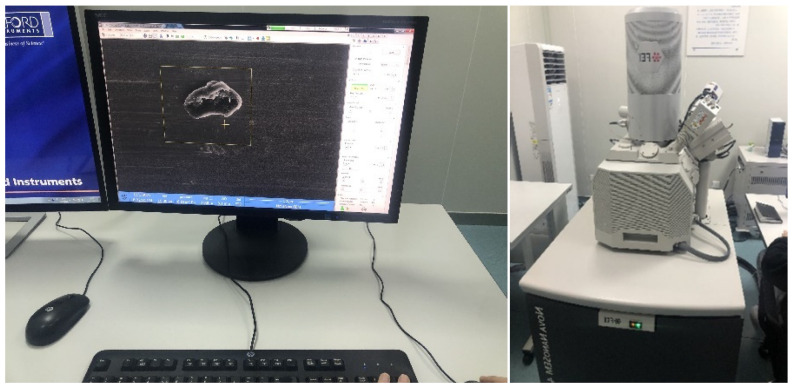
Field emission scanning electron microscope SEM.

**Figure 7 micromachines-14-00069-f007:**
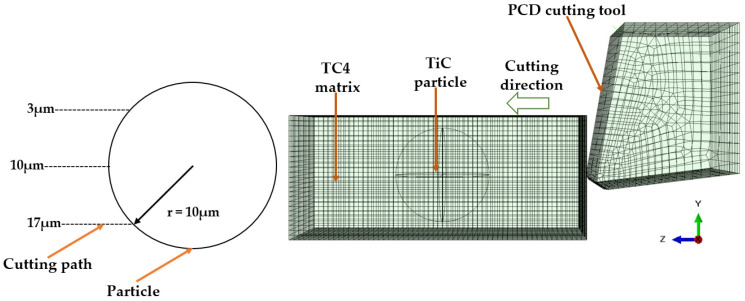
Single particle cutting simulation test design.

**Figure 8 micromachines-14-00069-f008:**
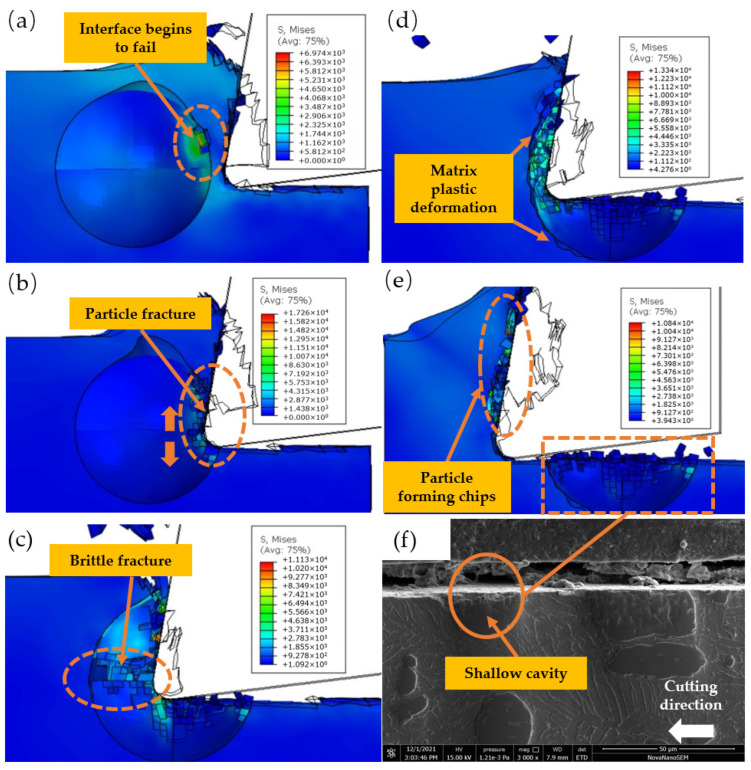
The simulated cutting process on the cutting path: (**a**) Cracking of TiC particles; (**b**) Tool contact with TiC particles; (**c**) Brittle Fracture of TiC Particles; (**d**) TC4 matrix damage; (**e**) Simulation results of brittle fracture of TiC particles; (**f**) Particle brittle fracture test results.

**Figure 9 micromachines-14-00069-f009:**
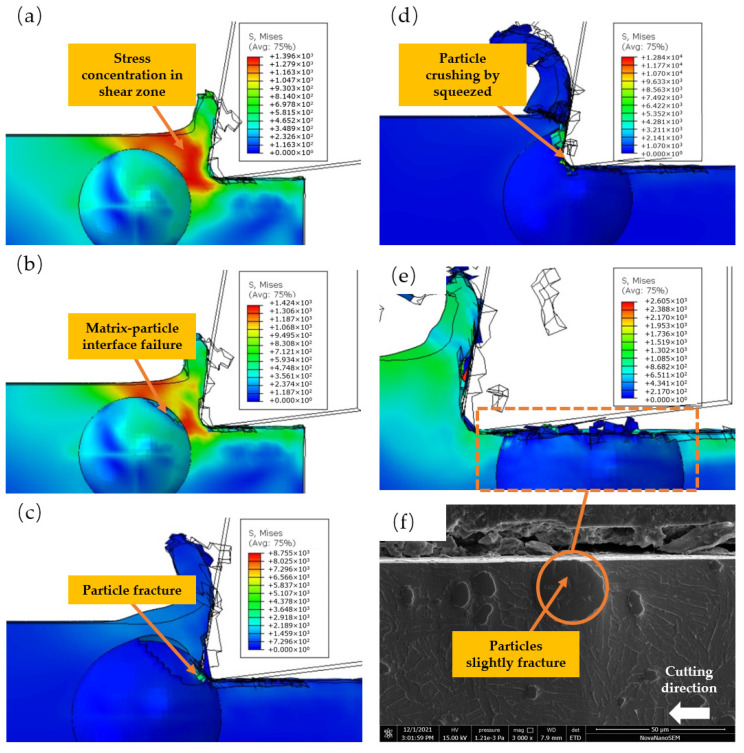
Simulated cutting process of particle below cutting path: (**a**) stress concentration near matrix-particle interface; (**b**) matrix-particle interface failure; (**c**) tool contact with TiC particle; (**d**) TiC particles are broken by tool extrusion; (**e**) simulation results of slightly fractured TiC particles; and (**f**) test results of slightly fractured TiC particles.

**Figure 10 micromachines-14-00069-f010:**
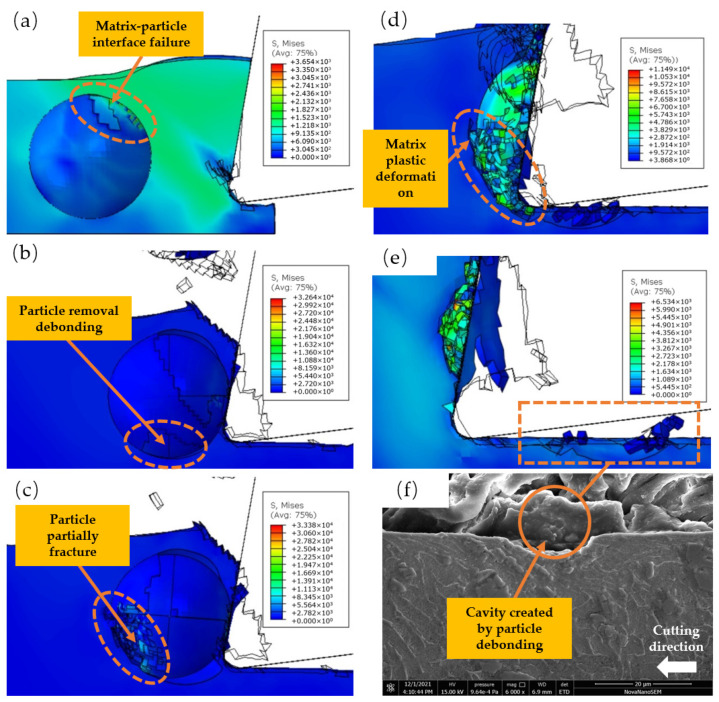
Simulation of cutting process of particles above cutting path: (**a**) matrix-particle interface failure above TiC particles; (**b**) TiC particles that are in contact with the tool; (**c**) TiC particle cleavage under tool extrusion; (**d**) TC4 matrix scratched by TiC particles; (**e**) simulation results of cavity generated by particle debonding; and (**f**) test results of the cavity produced by particle debonding.

**Figure 11 micromachines-14-00069-f011:**
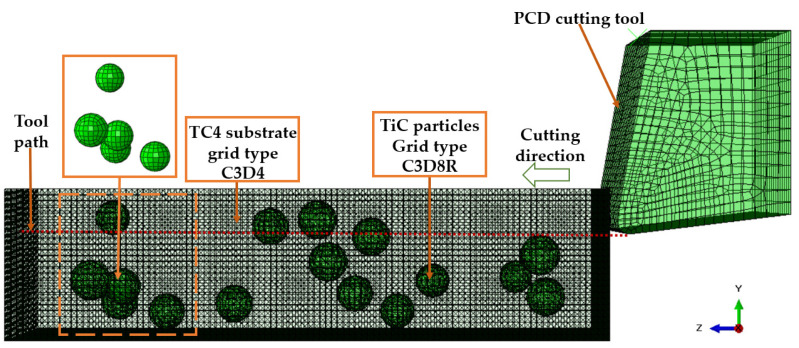
Simulation model of random distribution of enhanced particles in PTMCs.

**Figure 12 micromachines-14-00069-f012:**
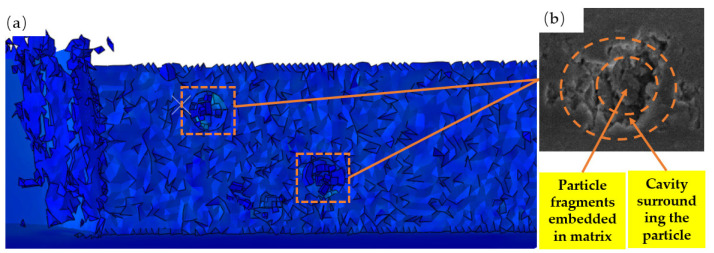
Machining surface shallow voids around intact particles: (**a**) simulated particles are removed to form a shallow cavity, and (**b**) the surface shallow voids around intact particles SEM image after turning to process.

**Figure 13 micromachines-14-00069-f013:**
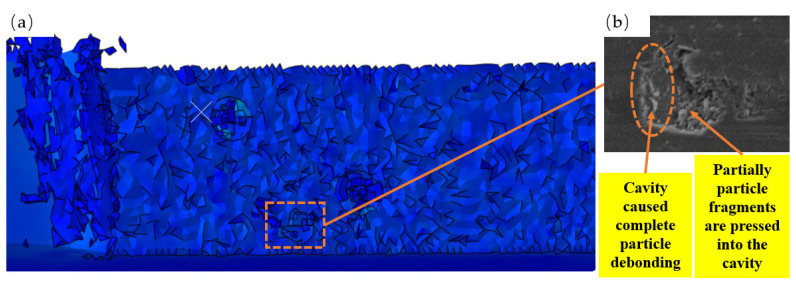
Machining surface cavity: (**a**) simulated particles are removed to form a cavity and (**b**) the surface cavity SEM image after turning to process.

**Figure 14 micromachines-14-00069-f014:**
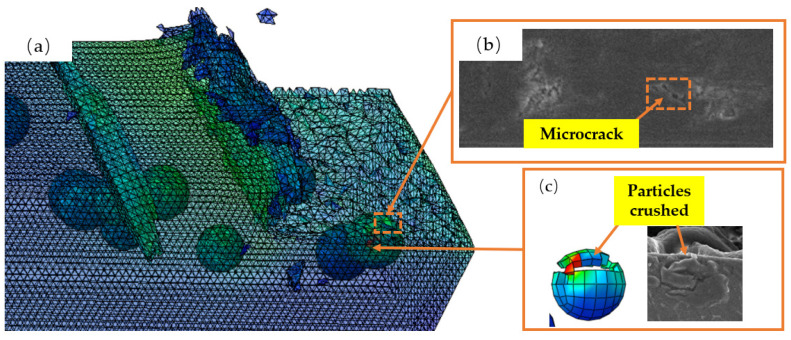
Machining surface micro-crack: (**a**) simulated particles are removed to form micro-crack; (**b**) the surface micro-crack SEM image after turning to process; (**c**) diagram of TiC particles crushed.

**Table 1 micromachines-14-00069-t001:** Material properties of TC4 [24].

Parameters	Values
*A* (Mpa)	1098
*B* (Mpa)	1092
*T_m_* (*°*C)	1560
*T_r_* (*°*C)	20
*C*	0.01
*n*	0.386
*m*	0.71

**Table 2 micromachines-14-00069-t002:** J-C shear failure parameters of TC4 titanium alloy [24].

Parameters	Values
*d* _1_	−0.09
*d* _2_	0.27
*d* _3_	0.48
*d* _4_	0.014
*d* _5_	3.87

**Table 3 micromachines-14-00069-t003:** Material properties of TiC [26].

Parameters	Values
Tensile strength (Mpa)	4390
Type I Fracture Energy of Materials (J/m^2^)	27
Maintaining factor	1
Cracking strain	0.0001
Crack opening strain	0.001

**Table 4 micromachines-14-00069-t004:** Physical and mechanical properties of titanium matrix composites [27].

Parameters	Values
Materials	TiC_p_/TC4 composites
Enhanced Phase Type	Particles
Enhanced phase content (%)	5
Modulus of elasticity/MPa	121
Yield strength/MPa	934.8
Tensile strength/MPa	1030.2
Enhanced phase average size (μm)	1.5~20
Hardness (HRC)	34~36
Thermal conductivity	5.708
Elongation at break (%)	9.3

**Table 5 micromachines-14-00069-t005:** PCD tool working angle and grain size.

Parameters	Values
Tool types	PCD
Front corner (°)	5
Rear corner (°)	8
Main declination (°)	45
Sub-deviation angle (°)	45
Blade Tilt (°)	4
Grain size (μm)	2&30
The radius of the tool-tip circle (mm)	0.8

**Table 6 micromachines-14-00069-t006:** Turning cutting parameters.

Parameters	Values
Enhanced phase content (%)	5
Cutting speed (m/min)	45
Feed rate (mm)	0.08
Cutting depth (mm)	0.5

## Data Availability

All data needed to evaluate the conclusions in the paper are present in the paper. Additional data related to this paper may be requested from the authors.

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
