# Peer review of "Simulation and Experimental Analysis of Surface Defects in Turning of TiCp/TC4 Composites"

_micromachines, 2022, doi:10.3390/mi14010069_

Round 1

Reviewer 1 Report

The study presented in this research is sound, and the results produced are interesting. But a revision is required, and after responding to the following remarks and revising the paper, the manuscript may be considered for publication.

1. Literature review needs to include several recent, relevant publications (high impact) highlighting their key findings. The current version only discussed general aspects while the review of each from several papers is necessary. You may provide a review summary table consisting of a column for the comments or key conclusions.

2. More recent relevant literature or similar work discussion is mandatory in the introduction section, which is missing in the Introduction. Authors are suggested to add one paragraph in the introduction section by discussing the recent progress and citing similar work.

3. The novelty of the work is missing in the introduction. Authors are suggested to include a separate paragraph discussing the novelty and importance of the present work.

4. Authors are suggested to include a literature review on the recent publication on composite materials based on the following references in the introduction section: DOIs: 10.1016/j.rinp.2018.06.010; 10.1088/2053-1591/ab22d8; 10.1016/j.rinp.2019.102264; 10.1080/10420150.2019.1606809.

5. Reduce the similarity. Check the attached similarity report.

6. Also, check the typos throughout the manuscript during revision submission.

Reviewer 2 Report

The manuscript is written concisely. Suggestions for improving the manuscript are as follows:

1. How did you choose the machining parameters? Why are they representative?

2. What is the coefficient of friction? Elaborate in detail.

3. What fixtures did you use? How much influence does the fixtures have on the results obtained?

4. Potential errors have not been analysed and discussed.

5. The discussion of the obtained results must be more intensive. The physics of the process, causes and effects must be discussed. The obtained results should be scientifically compared with the results of previous research.

6. Emphasize the possibilities of industrial application.

Reviewer 3 Report

The paper deals with the Simulation and Tests Analysis of Surface Defects in Turning of TiCp/TC4 Composites. 

According to the reviewer, the paper is worth publishing at micromachines Journal, but corrections are needed and then the paper can be accepted for publication in the journal.

While the authors have made considerable research effort, 

the presentation of the paper and the results must be proved. 

Additionally make the following corrections to the manuscript:

Comment 1

Line 63

Qi et al. [7] studied

The authors should replace (delete the "et al.")

Qi [7] studied

Line 85

Zhan C et al. [10] carried

The authors should replace (delete the "et al.")

Zhan C [10] carried

Comment 2

Line 120

as shown in Figure 1.

The authors must format the paper according to the journal's instructions.

Figures should be placed in the main text near to the first time they are cited.

Comment 3

Extended text editing

Line 161

the following formula.[14].

The authors should replace

the following formula [14].

Line 196 

2)is the

The authors should replace

2) is the

Line 212

numerically controlled machine tool,

Different text font size. 

Line 237

composites. [19]In order

The authors should replace

composites [19]. In order

Line 331

Figure 8a When the shear zone

The authors should check if the sentence is right.

Line 483

Matrix Composites. . chinese Journal of 

The authors should replace

Matrix Composites. Chinese Journal of 

Comment 4

The authors must explain with more details the equation 4 (σ1, σ2 and σ3 = ?)

and why the authors used the "where ?o is the tensile strength of TiC particles." (there is not ?o in the equation 4).

Comment 5

Line 206

The authors must give more details for experiment equipments (SEM: type, model).

Comment 6

Lines 215 and 451

The authors should format the paper:

Delete the space in the page end

Comment 7

The authors must give more results for the Turning cutting parameters (Cutting speed (m/min) 15/30/45/60/90/120/150 - not only for 45 m/min)

Comment 8

The authors must give more details for FEM simulations (number of elements, minimum length, .....)

Comment 9

Line 245

It's not so good to start the sub-section at the bottom of the page without using text.

The authors should format.

Lines 277 and 306

It's not so good to start the sub-section with Figure and without using text (First text, then Figure).

The authors should format.

Comment 10

Figure 6

The authors must add the units for the Von Mises stress (MPa?).

Comment 11

Increase the number of the reference papers including (primarily) from MDPI Journals.

Improove the quality of the reference papers.

More related literature should be included in introduction section about the FEM simulation. 

Please consider the following ones, which are all related with FEM simulation: 

(a) https://doi.org/10.3390/ma12162522

(b) https://doi.org/10.3390/met10030338

Round 2

Reviewer 2 Report

The manuscript has been corrected.

Author Response

Thank you for your comments and suggestions

Reviewer 3 Report

Comment 1

Extended text editing.

Line 10

produced., processing

All the ref. with two or more authors must be mention with et al.

For examble:

Line 46

Xiong [4]

The authors should replace

Xiong et al. [4]

Line 51

mm mm 

The authors should replace

mm

Line 80

Pramnik [15]

The authors should replace

Pramanik et al. [15]

Comment 2

Line 106

both local and international researchers

The authors comment the References in a very large percentage from local researchers. It would be preferable to refer to the Paper

with a more balanced rate.

Comment 3

Line 116

3D microstructure-based cutting simulation

Page 14 Figure 14 (not 12)

FEM simulation shows more in orthogonal cutting (and not turning).

Comment 4

The Table 4 must be accompanied on the same page as the Table's title.

The Table 6 must be accompanied on the same page as the Table's title.

Comment 5

Table 6

Cutting speed (m/min): only 45 (delete the other values, the authors did not give any results).

Comment 6

Lines 315 - 316

7000 and 10000 MPa

The authors must explain (give more details) the results.

The authors must give more details for temperature.

Comment 7

Line 374

the length, height and breadth of the model are set to 100m, 25 m, and 25 m, respectively.

The authors must check if the values are right (m).

Comment 8

Figure 11: 2 times!!!

Page 12 and Page 13

Figure 12: 2 times!!!

Page 13 and Page 14

Comment 9

The Section References must begin from Line 468.

Author Response

Thank you for your comments and suggestions. Please see the attachment

Round 3

Reviewer 3 Report

It is disappointing that the authors do not consider the reviewer's comments.

New Comment 1 (Previous Comment 1)

Line 80 (new line 320)

Pramnik [15]

The authors should replace

Pramanik et al. [15]

The name of the author for the ref [15] is Pramanik (no Pramnik as the authors of this paper mention again and again).

New Comment 2 (Previous Comment 1)

Extended text editing.

Line 920

Figure 10a Matrix-particle interface ruptures

There is not a sentence and the authors must rephrase.

New Comment 3 (Previous Comment 8)

Figure 11: 2 times!!!

Page 12 and Page 13

Author Response

I am very sorry for my carelessness in the previous rewriting process. This time I have checked my manuscript several times and used the 'track changes' feature during the revision process. Finally, thank you again for your always exceptionally detailed comments and suggestions. Please see the attachment.
